# Evaluation of photoreceptor features in retinitis pigmentosa with cystoid macular edema by using an adaptive optics fundus camera

**Shohei Kitahata**[1,2,3]*, **Kiyoko Gocho**[1,4]*, **Naohiro Motozawa**[5], **Satoshi Yokota**[1,3],
**Midori Yamamoto**[1], **Akiko Maeda**[1], **Yasuhiko Hirami**[1,3], **Yasuo Kurimoto**[1,3],
**Kazuaki Kadonosono**[2], **Masayo Takahashi**[1,5,6]

1 Kobe City Eye Hospital, Chuo-ku, Kobe, Japan, 2 Department of Ophthalmology and Micro-Technology, Yokohama City University, Minami-ku, Yokohama, Japan, 3 Department of Ophthalmology, Kobe City Medical Center General Hospital, Chuo-ku, Kobe, Japan, 4 Inserm and Quinze-Vingts National Ophthalmology Hospital, Paris, France, 5 RIKEN Center for Biosystems Dynamics Research, Chuo-ku, Kobe, Japan, 6 Vision Care Inc., Chuo-ku, Kobe, Japan

* kitahata@yokohama-cu.ac.jp (SK); kknaka17@gmail.com (KG)

**Data Availability Statement:** All relevant data are within the paper and its Supporting Information files.

## Abstract

### Objective

Cystoid macular edema (CME) in retinitis pigmentosa (RP) is an important complication causing visual dysfunction. We investigated the effect of CME on photoreceptors in RP patients with previous or current CME, using an adaptive optics (AO) fundus camera.

### Methods

We retrospectively observed the CME and ellipsoid zone (EZ) length (average of horizontal and vertical sections) by optical coherence tomography. The density and regularity of the arrangement of photoreceptor cells (Voronoi analysis) were examined at four points around 1.5° from superior to inferior and temporal to nasal. We also performed a multivariate analysis using CME duration, central macular thickness and transversal length of CME.

### Results

We evaluated 18 patients with previous or current CME (18 eyes; age, 48.7 ± 15.6 years) and 24 patients without previous or current CME (24 eyes; age, 46.0 ± 14.5 years). There were no significant differences in age, logMAR visual acuity, or EZ length. In groups with and without CME, cell density was 11967 ± 3148 and 16239 ± 2935 cells/mm$^2$, and sequence regularity was 85.5 ± 3.4% and 88.5 ± 2.8%, respectively; both parameters were significantly different. The correlation between photoreceptor density and age was more negative in group with CME. The CME group tended toward greater reductions in duration of CME.

**Funding:** The authors received no specific funding for this work.

**Competing interests:** The authors have declared that no competing interests exist.

## Conclusion

Complications of CME in RP patients may lead to a decrease in photoreceptor density and regularity. Additionally, a longer duration of CME may result in a greater reduction in photoreceptor density.

## Introduction

Retinitis pigmentosa (RP) is an intractable disease affecting 1 in 3000–5000 people and is characterized by progressive degeneration of photoreceptors and/or the retinal pigment epithelium (RPE) [1]. RP is the result of a mutation in one of more than 260 genes that are associated with various processes, such as the synthesis of peptides involved in the visual cycle, the conversion of glucose to adenosine triphosphate, and the removal of metabolic waste products [2, 3]. Typical RP involves progressive bilateral degeneration of the rod and cone photoreceptors, in which the first symptom is night blindness due to rod cell death followed by progressive peripheral visual field defects. However, central vision is generally preserved for several decades because of the maintenance of cone photoreceptors [4]. Cystoid macular edema (CME) is a common yet critical complication of central vision impairment in such patients [5]. The prevalence of CME among patients with RP has been cited as 11–20% based on fluorescein angiography (FA) and fundus examination, while the diagnosis rate using optical coherence tomography (OCT) has been cited as 38–49% [6–9]. Thus, the possibility and risk of CME should be considered in the early stage of RP and evaluated using OCT. The CME caused by inflammation and diabetes results in diminished photoreceptor function, indicating directional sensitivity and visual pigment density [10]. A decrease in cone density has recently been noted in the presence of edema related to the macular telangiectasia type 2 and the diabetic macular edema in studies using adaptive optics (AO) fundus imaging [11, 12]. Researchers have highlighted the potential value of AO for the non-invasive examination of photoreceptors at a resolution of up to 2 μm given its ability to reduce the effect of optical aberrations, yielding findings that cannot be obtained using conventional retinal imaging techniques [13]. The reflectance of the cone photoreceptor mosaic in AO images using a flood illuminated retinal fundus camera (rtx1 TM, Orsay, France) is substantially considered as a reflection of the outer segments of the photoreceptors or the interdigitation zone (IZ) in OCT images [14]. Unlike measurements obtained using the AO-scanning laser ophthalmoscopy (AO-SLO) system which includes a confocal split-detection device, AO examination using an rtx1 device is considered to measure the visibility of the outer segment rather than the exitance of photoreceptors, meaning that it may be possible to evaluate visual function indirectly using this method [15]. Little has been reported on the detailed relationships between the cone density and CME in patients with RP. As such, methods that allow for visualization of the cone outer segments and photoreceptors are desirable. In this study, we used OCT and AO to investigate the effect of CME on photoreceptors in patients with RP and analyzed the effects of edema-related factors.

## Methods

### Patients

This retrospective cohort study included the eyes of patients seen in RP clinics at Kobe City Eye Hospital, Kobe, Japan, from 2015 to 2021. The study protocol was conducted in

accordance with the tenets of the Declaration of Helsinki. We obtained written informed consent from all patients for the analysis of genetic testing, which had been approved by the Institutional Review Board of Kobe City Medical Center General Hospital (Protocol no. E19002 and Permit no. ezh200901, 04.09.2020). Patient data were obtained from Kobe City Eye Hospital and Kobe City Medical Center General Hospital. The temporal interval during which we procured patient data for the purpose of data aggregation spanned from April of the year 2021 to March of the year 2022. The diagnosis of RP was based on the patient's history of night blindness, family history, and clinical evaluations, including visual field constriction, bone-spicule-like pigment clumping, presence of central perimacular hyperautofluorescent rings, molecular genetic testing, and full-field electroretinogram resluts suggestive of rod-cone dystrophy. CME was diagnosed based on the results of spectral domain (SD) OCT (Spectralis; Heidelberg Engineering, Heidelberg, Germany). The criterion for CME with RP was the presence of visible intraretinal cystoid spaces in the macula, confirmed using horizontal and/or vertical OCT measurements. The control eyes were those of patients with RP who had no history of CME based on OCT examination. In the comparison groups, only the right eye was used for observation.

## Inclusion/exclusion criteria

To acquire good-quality AO images, we included patients who were able to fixate on the machine's target. Images that showed a loss of focus, blink lines, or motion artifacts were not included in this study. In the first selection, 48 patients with RP patients were included in the study; however, images adequate for analyzing cone photoreceptor cells were obtained only in 42 RP patients. The exclusion criteria were the presence of other degenerative diseases such as cone-rod dystrophy, cone dystrophy, Bietti crystalline retinopathy, retinal inflammation diseases, or autoimmune retinopathy.

## Measurements of clinical parameters

Each patient underwent a comprehensive eye examination at least once, which included measurements of decimal best-corrected visual acuity (BCVA), SD-OCT, axial length (IOLMaster; Carl Zeiss Meditec, Jena, Germany), and cone photoreceptor analysis using an AO fundus camera (rtx1TM, Imagine Eyes, Orsay, France). The rtx1 is an AO flood-illuminated retinal camera, and its principles have been reported in detail [16]. BCVA was assessed with full subjective refraction using a Landolt C-chart and converted to the logarithm of the minimal angle of resolution units (logMAR) for statistical analysis. For very low vision, such as counting fingers (CF), hand movement (HM), and light perception (LP), category evaluation was used for analysis [17, 18].

## AO analysis

For the cone photoreceptor analysis in this study, we used AO images acquired using the rtx1 device and analyzed them with software provided by the manufacturer, as shown in Fig 1 (AOdetect mosaic, version 0.1; Imagine Eyes). For each photoreceptor analysis examination, we captured the scans of the four perifoveal areas of the retina, as the regions of interest (ROI): superior, inferior, temporal, and nasal, 1.5° to 2° from the center of the fovea with a standardized sampling window of $62 \times 62$ μm sampling window size (for an eye axis of 24 mm) [19]. We performed measurements using AOdetect and used the axial length from each participant to correct the measurements. The software calculated the local cell density (in cells per square millimeter) and array regularity (%) based on Voronoi analysis [20]. The data from the four ROIs were averaged and used for all analyses in this study. In order to avoid the influence of

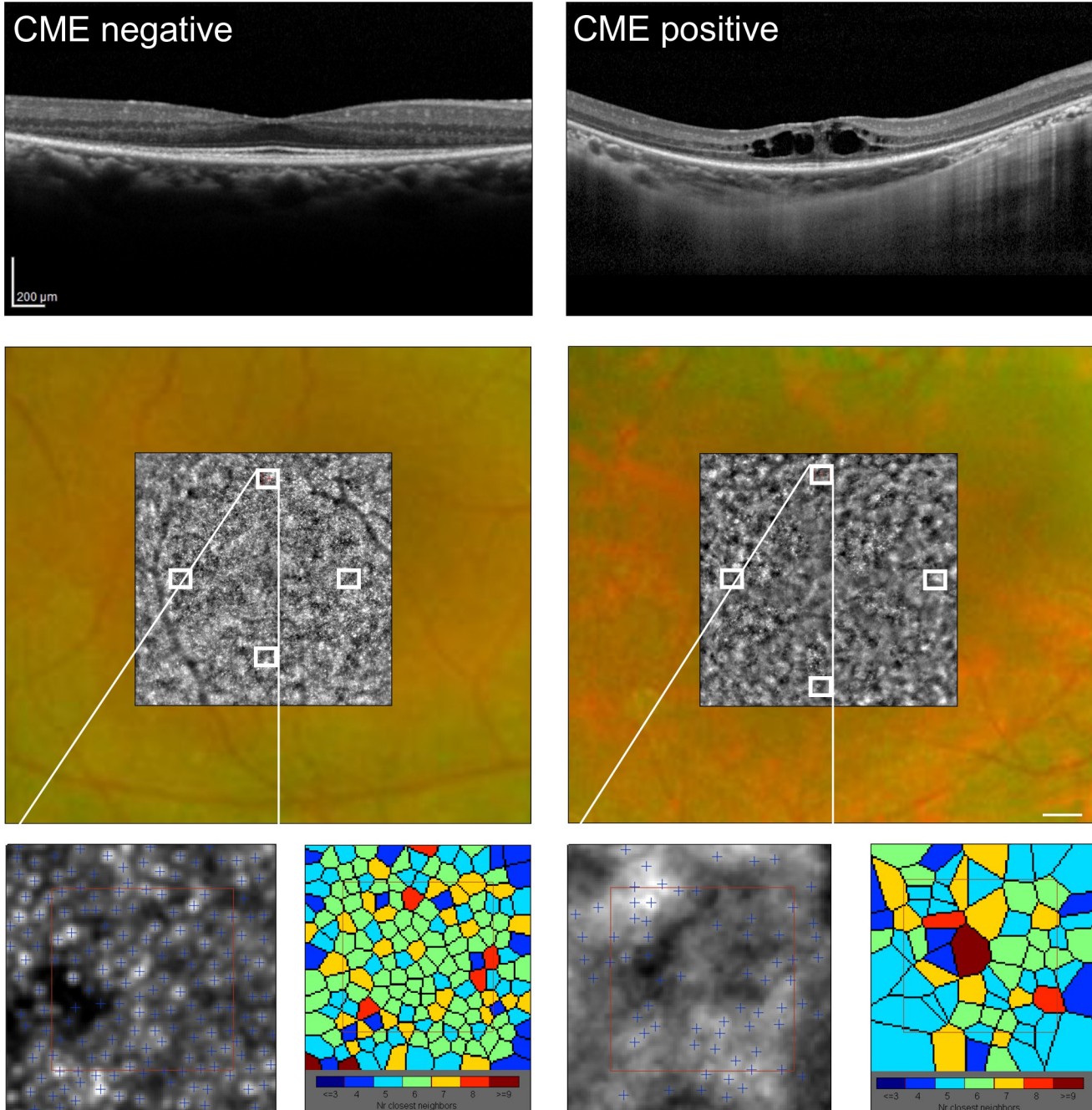

**Fig 1. Representative adaptive optics (AO) image of the right eye from CME-negative/positive patients.** The upper images are OCT horizontal scan images. The middle images show fundus photographs with an AO image superimposed on the corresponding area. The lower images were processed using the analysis provided by the AO detect software. Photoreceptors were evaluated individually and are shown in the Voronoi diagram. Scale bar = 200 μm.

CME on the measurement results of the AO examination, the ROIs were determined in the same procedure for the CME positive and negative group, using the peak density method [21]. For the CME positive group, the measurements were obtained from the area in which CME was not seen in the OCT examination. It is crucial to note that, due to the utilization of the peak density method for measurements, any minor positional deviations in the measurement

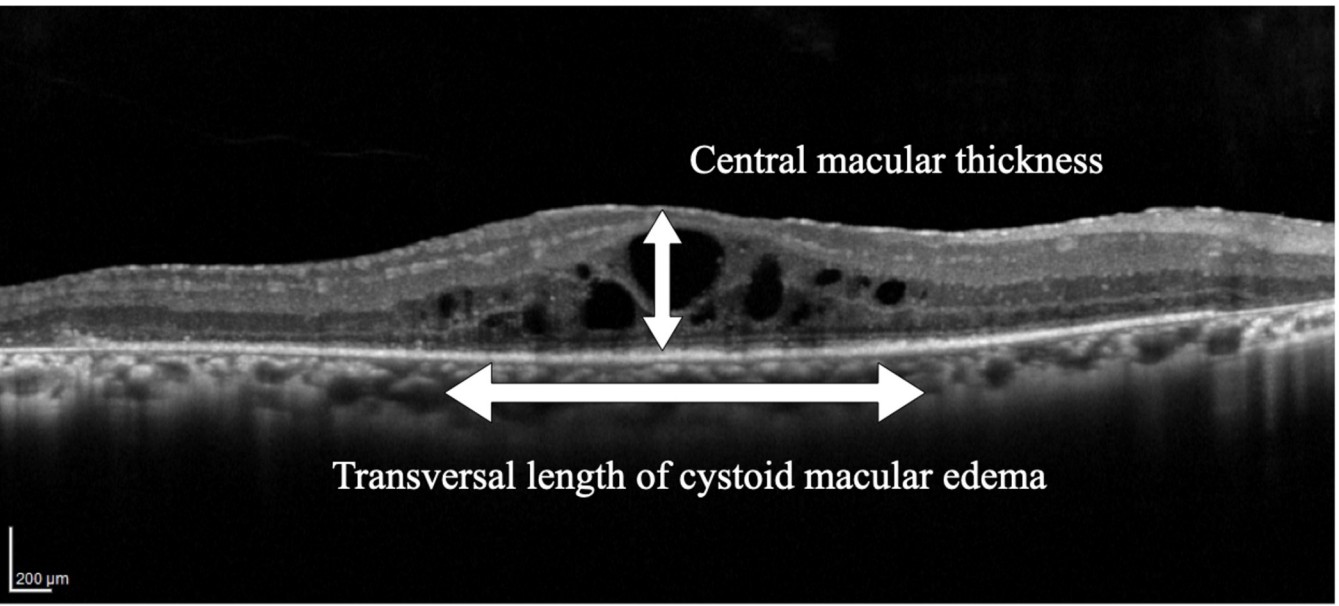

**Fig 2. Central macular thickness and transverse length of cystoid macular edema are identified as white arrows.**

points are considered inconsequential [21]. To prevent miscounting the hyper/hypo-reflective spots in the AO images, the cones labeled by the automated algorithm were visually examined by two retinal specialists. To confirm the validity of the measurements, we compared our results with previously reported data [22].

## OCT analysis

A Spectralis Viewing Module (V.6.0.14.0, Heidelberg Engineering) and an OCT viewer (V.2.14; Topcon, Tokyo, Japan) were used to measure each parameter described in Fig 2: retinal thickness at the fovea, transverse length of CME (TLC), transverse length of preserved EZ, and transverse length of the preserved IZ. Central foveal thickness (CFT) was defined as the distance between the vitreoretinal interface and the inner border of the RPE, and we regarded it as the average of vertical and horizontal foveal thickness values. The transverse lengths of the EZ and IZ were measured in the same manner. The duration of CME was determined from the time of the initial diagnosis of edema until the last visit when it was still detected in the SD-OCT.

## Genetic testing

All patients underwent DNA sequencing for a panel of 39 (13 patients) or 50 (29 patients) genes (Table 1) known to be associated with inherited retinal diseases, which were selected based on previous reports [23–25]. Genomic DNA was extracted from the participants' peripheral lymphocytes. DNA libraries were prepared using the KAPA HyperPlus Kit (KAPA Biosystems). Target regions were examined using a SureSelect XT DNA Capture Custom Kit (Agilent Technologies). Then, the targeted libraries were sequenced on an Illumina NextSeq (Next Seq 500 System; Illumina) that generated paired-end reads of 75 nucleotides at 1000× coverage. The findings related to detected variants and the molecular diagnosis of each patient were reviewed by a diverse team of professionals, including ophthalmologists, clinical geneticists, optometrists, nurses, researchers, and genetic counselors, all in accordance with the

**Table 1. List of genes in the target capture panel.**

| | | | | |
|---|---|---|---|---|
| ABCA4 | BEST1 | BBS1* | C2orf71 | CEP290* |
| CDH23* | CDHR1* | CHM* | CNGA1 | CNGB1 |
| CNGB3 | CRB1 | CRX | CYP4V2 | EYS |
| FAM161A* | GPR98* | GUCA1A* | GUCY2D | IMPDH1 |
| IMPG2 | KLHL7* | LRAT | MAK | MERTK |
| MYO7A* | NR2E3 | NRL | PCDH15* | PDE6B |
| PRCD | PROM1 | PRPF31 | PRPF6 | PRPH2 |
| RDH5 | RDH12 | RHO | ROM1 | RP1 |
| RP1L1 | RP2 | RP9 | RPE65 | RPGR |
| RS1* | SNRNP200 | TOPORS | TULP1 | USH2A |

*indicates genes involved only in the 50-genes panel. In the case of the 39 genes, GUCA1B was analyzed in addition to the above.

criteria and guidelines endorsed by the American College of Medical Genetics and Genomics and the Association for Molecular Pathology [26].

## Statistical analyses

The values of each parameter are presented as mean ± standard deviation. Statistical analysis was performed using R version 3.1.3 (R Foundation, Vienna, Austria). Comparisons between groups were performed using Welch's t-test. The generalized estimating equation (GEE) method was used to analyze each correlation, and multivariate analysis was performed between photoreceptor characteristics and each of the following parameters: CFT, TLC and CME duration. In the correlation analysis, Pearson's correlation coefficient values were used for parametric analysis, and Spearman's rank correlation was used for non-parametric analyses. Results were considered significant at $p < 0.05$. Each measurement was evaluated by two ophthalmologists. The intraclass correlation coefficients (ICCs) for EZ/IZ length, CFT, TLC, and photoreceptor density/regularity are summarized in Table 2.

## Results

Data for a total of 42 eyes of 42 patients, including 18 eyes of 18 patients with current or previous CME (CME-positive group; 13 eyes in current, 5 eyes in previous), and 24 eyes of 24 patients without a history of CME (CME-negative group), are summarized in Table 3. The genetic mutation of each patient was described in the S1 Table. Twelve of the eighteen patients (66.7%) had bilateral CME. The mutations of each gene in all patients in the CME-positive and -negative groups are shown in Fig 3 (including binocular onset; S1 Fig). In this study, no significant differences were observed for gene mutations and CME complication rates (Fisher's

**Table 2. Intraclass correlation coefficients for each measurement.**

| Variable | ICC | 95% confidence interval |
|---|---|---|
| EZ | 0.815 | $0.701 < ICC < 0.843$ |
| IZ | 0.843 | $0.728 < ICC < 0.899$ |
| Central macular thickness | 0.867 | $0.761 < ICC < 0.928$ |
| Transversal length of cystoid macular edema | 0.856 | $0.768 < ICC < 0.916$ |
| Photoreceptor density | 0.769 | $0.668 < ICC < 0.846$ |
| Photoreceptor regularity | 0.795 | $0.688 < ICC < 0.878$ |

ICC, intraclass correlation coefficient; EZ, ellipsoid zone; IZ, interdigitation zone.

**Table 3. Clinical data for patients with retinitis pigmentosa with and without a history of cystoid macular edema.**

| Characteristic | CME-positive | CME-negative | P value |
|---|---|---|---|
| Patient age (year) | 48.7 ± 15.6 | 46.0 ± 14.5 | 0.56 |
| Sex | | | |
| Male (%) | 2 (11.1%) | 10 (41.7%) | |
| Female (%) | 16 (88.9%) | 14 (58.3%) | |
| Laterality | | | |
| Right (%) | 12 (66.7%) | 24 (100%) | |
| Left (%) | 6 (33.3%) | 0 | |
| logMAR visual acuity | 0.13 ± 0.25 | 0.21 ± 0.41 | 0.43 |
| Axial length (mm) | 23.81 ± 1.20 | 24.52 ± 1.30 | 0.08 |
| Baseline lens status | | | |
| Phakic | 15 (83.3%) | 19 (79.2%) | |
| Pseudophakic | 3 (17.7%) | 5 (20.8%) | |
| Ellipsoid zone (μm) | 1827 ± 1039 | 1915 ± 1830 | 0.86 |
| Interdigitation zone (μm) | 489 ± 587 | 771 ± 960 | 0.28 |
| Density (cells/mm$^2$) | 11967 ± 3148 | 16239 ± 2935 | < 0.01 |
| Regularity (%) | 85.5 ± 3.7 | 88.5 ±2.8 | < 0.01 |
| Gene | | | |
| EYS | 3 (16.7%) | 13 (54.2%) | |
| MERTK | 0 | 1 (4.2%) | |
| PDE6B | 3 (16.7%) | 0 | |
| PRCD | 0 | 1 (4.2%) | |
| PRPF31 | 4 (22.2%) | 0 | |
| RHO | 2 (11.1%) | 1 (4.2%) | |
| RP-1 | 0 | 1 (4.2%) | |
| RPGR | 0 | 1 (4.2%) | |
| USH2A | 2 (11.1%) | 2 (8.3%) | |
| NA | 4 (22.2%) | 4 (16.7%) | |

Values are presented as number (%) or mean ± standard deviation.

CME, cystoid macular edema; logMAR, logarithm of the minimum angle of resolution; VA, visual acuity.

exact test; p = 0.303). Since the non-edema group included one eye per case while the edema group included two eyes per case for those with binocular onset, we used the eye with the shorter axial length to avoid artificiality. The two groups exhibited no significant differences in age, logMAR visual acuity, axial length, or EZ/IZ length (Table 3). In the CME-positive and -negative groups, cell density values were 11967 ± 3148 and 16239 ± 2935 cells/mm$^2$, and sequence regularity rates were 85.5 ± 3.7% and 88.5 ± 2.8%, respectively, indicating significant differences in photoreceptor density (p < 0.01, Welch's t-test) and regularity (p < 0.01, Welch's t-test) (Fig 4). Two eyes of one case are also shown in S2 and S3 Figs, although significant differences could not be tested because they were not completely independent samples.

In the analysis of the correlation between photoreceptor cells and EZ, a significant correlation between photoreceptor cells and EZ was observed in the CME-negative group (p < 0.01, Spearman's rank correlation), but not in the CME-positive group (p = 0.10, Spearman's rank correlation) (Fig 5A). The same tendency was observed for photoreceptor cells and the IZ: A correlation between photoreceptor cells and the IZ was observed in the CME-negative group (p < 0.01, Spearman's rank correlation), but not in the CME-positive group (p = 0.32, Spearman's rank correlation) (Fig 5B).

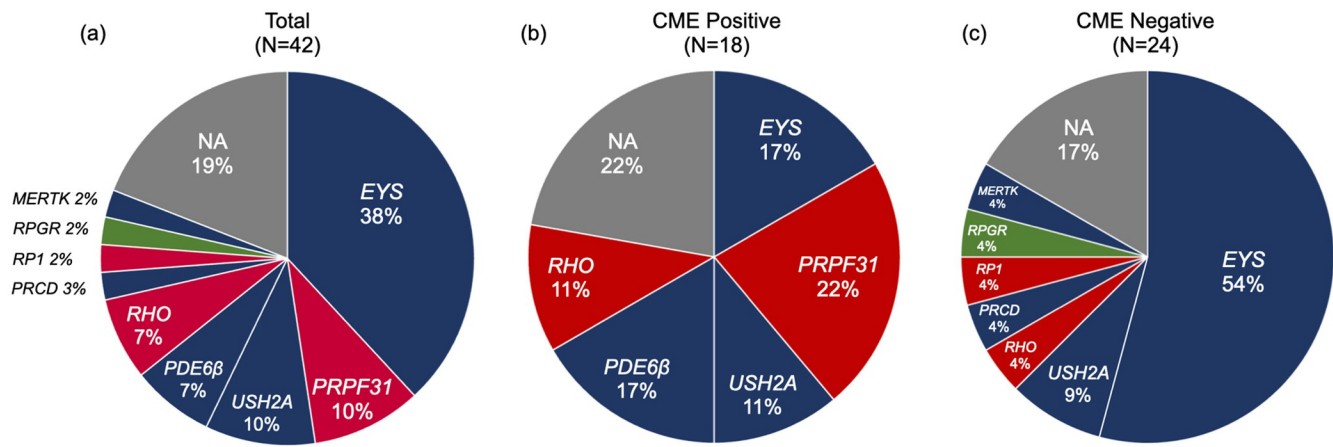

**Fig 3. Prevalence of gene mutation in retinitis pigmentosa patients.** (a) Total 42 eyes of 42 patients. (b) CME-positive group consisting of 18 eyes of 18 patients.(c) CME-negative group consisting of 24 eyes of 24 patients.

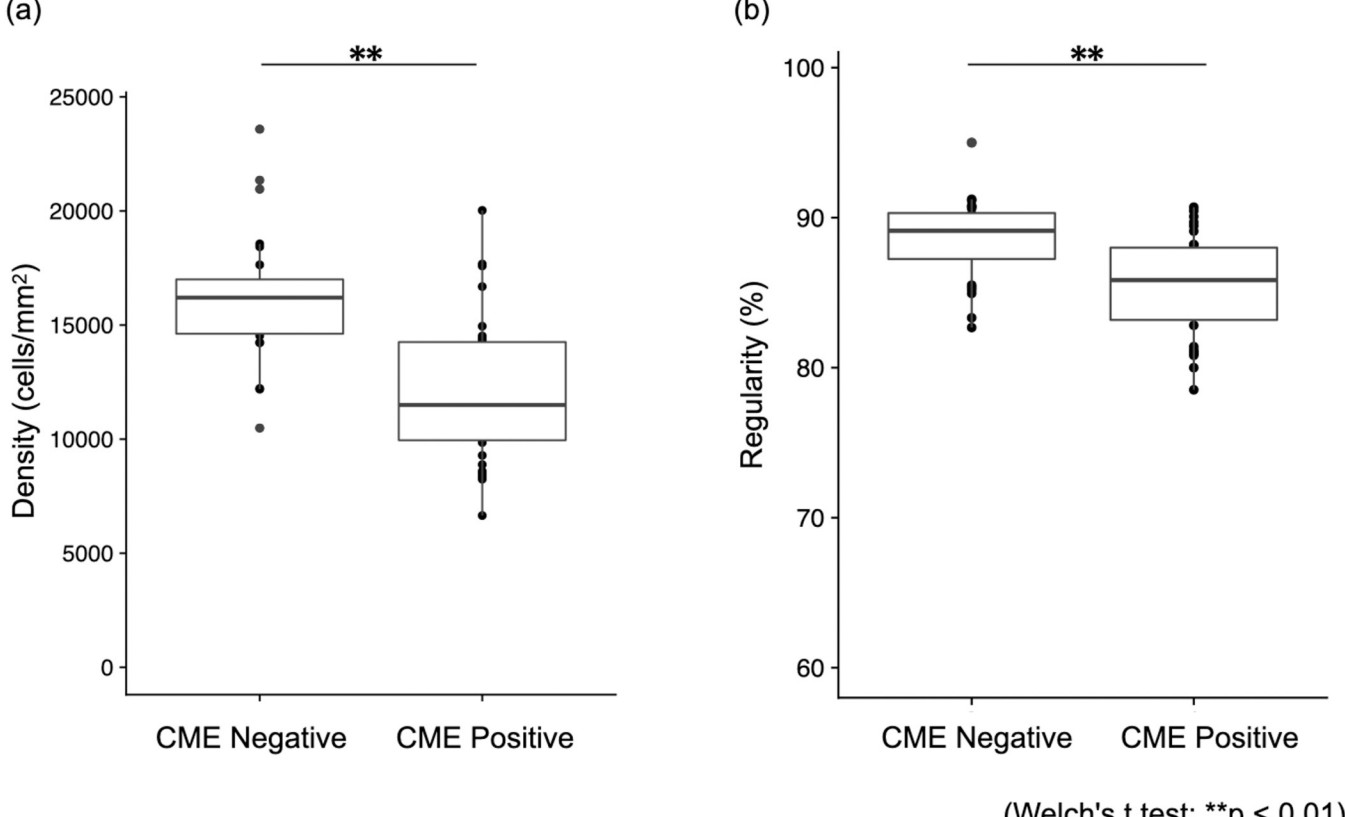

**Fig 4. Box-plot comparison of photoreceptors in groups with and without CME.** (a) Photoreceptor density. (b) Photoreceptor regularity.

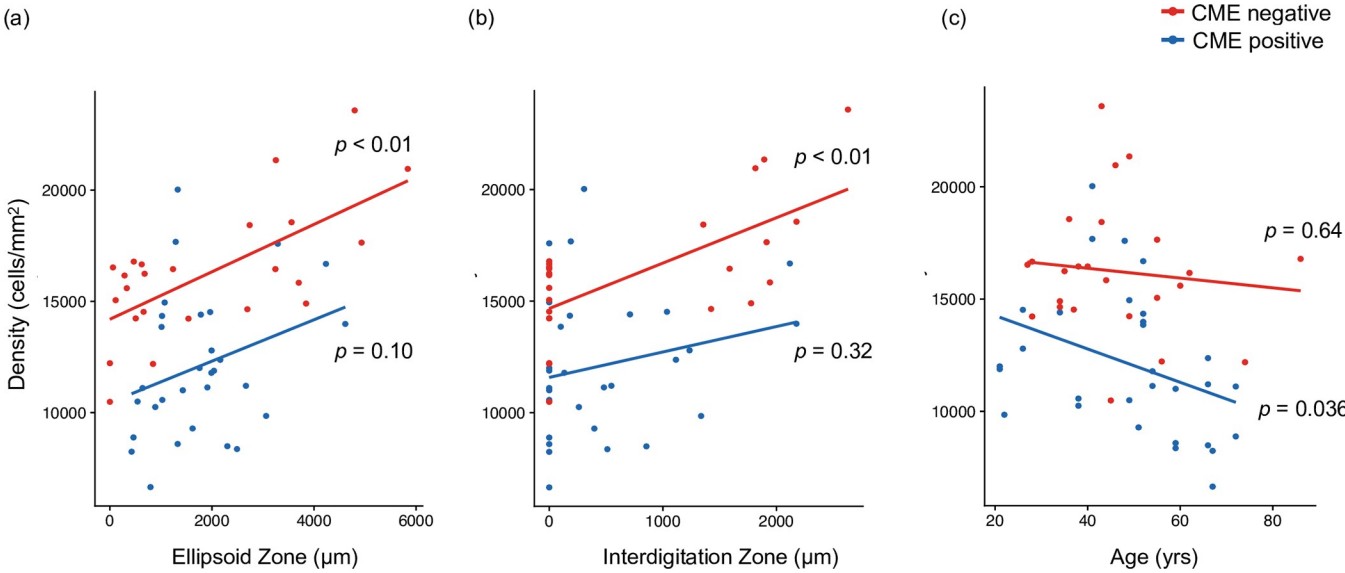

**Fig 5. Comparison of the correlation between photoreceptors and EZ/IZ/Age in the groups with and without CME.** (a) Correlation between photoreceptor density and EZ. In the analysis of the correlation between photoreceptor cells and EZ, a significant correlation between photoreceptor cells and EZ was observed in the CME-negative group (p < 0.01, Spearman's rank correlation), but not in the CME-positive group (p = 0.10, Spearman's rank correlation). (b) Correlation between photoreceptor density and IZ. A correlation between photoreceptor cells and the IZ was observed in the CME-negative group (p < 0.01, Spearman's rank correlation), but not in the CME-positive group (p = 0.32, Spearman's rank correlation). (c) Correlation between photoreceptor density and age. In the CME-negative group, we observed no correlation between photoreceptor density and age (p = 0.64, Pearson's correlation), while a negative correlation between photoreceptor density and age was observed in the CME-positive group (p = 0.036, respectively, Pearson's correlation).

Next, we examined correlations with patient age. In the CME-negative group, we observed no correlation between photoreceptor density and age (p = 0.64, Pearson's correlation), while a negative correlation between photoreceptor density and age was observed in the CME-positive group (p = 0.036, Pearson's correlation) (Fig 5C).

We observed significant negative correlations of visual acuity with the EZ (p < 0.01, Spearman's rank correlation), IZ (p < 0.01, Spearman's rank correlation), and photoreceptor density (p = 0.023, Spearman's rank correlation) in the CME-negative group (Fig 6A–6C), whereas there were no significant correlations for any items in the CME-positive group (EZ; p = 0.68, IZ; p = 0.63, density; p = 0.62).

Furthermore, in the CME-positive group, multivariate analysis was performed using macular edema-related items. Density decreased with age and increasing duration of CME (p < 0.01/ p = 0.041, generalized estimation equation) (Table 4). In contrast, in the CME-negative group, multivariate analysis using age revealed no significant differences.

## Discussion

We utilized AO technology to investigate the impact of CME on photoreceptors in patients with RP, as well as the influence of CME related items on the observed effects. Previous studies conducted using AO fundus examination have reported decreased cell density in patients with RP when compared with levels observed in healthy controls [27]. There was a significant decrease in photoreceptor density in the RP group in this study compared to the report of normal group measured by rtx1 [22]. Furthermore, when comparing the RP disease groups with and without CME complications, photoreceptor density was significantly decreased in CME positive group. Although the length of the EZ/IZ, which reflects the progression of RP, did not significantly differ between the groups, a history of CME was associated with photoreceptor

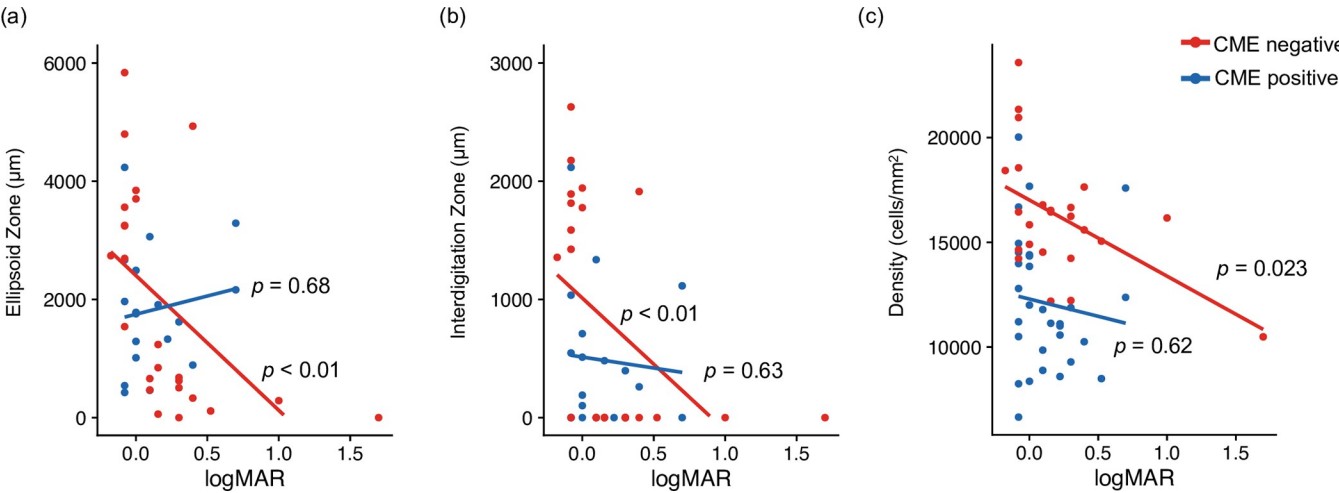

**Fig 6. Comparison of the correlations among EZ/IZ, photoreceptor density, and visual acuity in groups with and without CME.** (a) Correlation between EZ and logMAR. There was significant negative correlation of visual acuity with the EZ (p < 0.01, Spearman's rank correlation) in the CME-negative group. Whereas there was no significant correlation in the CME-positive group (p = 0.68, Spearman's rank correlation). (b) Correlation between IZ and logMAR. There was significant negative correlation of visual acuity with the IZ (p < 0.01, Spearman's rank correlation) in the CME-negative group. Whereas there was no significant correlation in the CME-positive group (p = 0.63, Spearman's rank correlation). (c) Correlation between photoreceptor density and logMAR. There was significant negative correlation of visual acuity with the photoreceptor density (p = 0.023, Spearman's rank correlation) in the CME-negative group. Whereas there was no significant correlation in the CME-positive group (p = 0.62, Spearman's rank correlation).

loss. To our knowledge, this study is the first to report an analysis of discernible photoreceptors utilizing rtx1 in patients with RP complicated by CME.

Although the effects of CME on photoreceptors have not been established to date, some studies have reported a relationship between macular edema and photoreceptor characteristics in the context of various diseases. Lammer et al. reported that decreased regularity of the cone arrangement was consistently associated with the presence of diabetic macular edema [11]. Jacob et al. reported that macular cone density decreased even outside the area of telangiectasia in patients with macular telangiectasia type 2, despite preservation of the EZ [12]. Voronoi and nearest neighbor analyses indicated that the branch retinal vein occlusion (BRVO)-affected side of the retina exhibited irregular shapes in the cone mosaic pattern. Even after recovery from macular edema-related BRVO, the regularity of the spatial arrangement of the cone mosaic was disrupted [28]. Studies assessing the effects of diabetic macular edema on photoreceptors have suggested that swelling of retinal cells anterior to photoreceptors can physically distort retinal structures and cause changes in the regularity and spacing of cone mosaics [11]. CME may also be caused by RPE impairment, especially transepithelial transport capacity, which may in turn damage photoreceptors [29]. As shown in Fig 5, EZ/IZ length was positively correlated with photoreceptors in the CME-negative group but not in the CME-positive

**Table 4. Multivariate analysis of the relationship between measurement parameters and photoreceptor densities in the CME-positive group.**

| Variable | Coefficients | SE | P-value |
|---|---|---|---|
| Maximum macular thickness | 3.402 | 10.412 | 0.74384 |
| Maximum transversal length of CME | -0.074 | 1.003 | 0.94117 |
| Duration of CME | -93.204 | 45.619 | 0.04104 |
| Age | -163.767 | 41.062 | 0.0044 |

SE, standard error; CME, cystoid macular edema.

group. A significant correlation has been reported between the annual progression of retinal sensitivity and EZ/IZ length in patients with RP [30], and EZ/IZ length measured based on reflectivity in en-face OCT has been suggested to be correlated with photoreceptor density [31]. Although it is difficult to be sure because of the small number of patients, the reduction in photoreceptor density in the CME-positive group, regardless of EZ/IZ preservation, might be due to CME. The intriguing aspect lies in the potential of AO imaging to identify photoreceptor loss that remains invisible when observing EZ/IZ shortening.

In the CME-positive group, photoreceptor density decreased with age. The absence of a reduction in photoreceptor cells in the non-CME may indicate that RP degeneration occurred from the periphery and did not affect the center of the measurement range. Aging also does not significantly change cone density in the central region [32]. Based on our findings, a history of even one previous CME episode may be associated with persistent effects on photoreceptors, even if the edema eventually abates, consistent with findings stating that the effects of BRVO persist even after resolution of edema [28].

Next, we examined relationships with visual acuity, as shown in Fig 6. Hagiwara et al. reported that EZ/IZ length was significantly correlated with visual acuity. They also reported that the lengths were correlated with each other, with impairments first affecting the IZ, followed by the EZ [33]. Our results were similar, suggesting that IZ disruption appears first. In addition, while visual acuity exhibited a negative correlation with the EZ/IZ in the CME-negative group, no such correlation was observed in the CME-positive group. The same result was observed for photoreceptors, suggesting that CME distorted the original relationships among EZ/IZ, photoreceptors, and visual acuity. There were also no significant between-group differences in visual acuity or EZ/IZ length; however, there was a significant difference in photoreceptor density. Kim et al. reported that CME increased the risk of EZ/IZ disruption in patients with RP, although the presence of CME was not correlated with visual acuity [34]. Thus, CME may disrupt the EZ/IZ structure, resulting in impairments in retinal function.

Concerning edema-related factors such as duration of CME, CFT, and TLC, a correlation was only observed between the duration of CME and the decline in photoreceptor density. The underlying mechanisms of CME in patients with RP remain incompletely elucidated and are believed to be multifactorial [35, 36]. These encompass Müller cell hypertrophy and its paracrine influences, impairment of the RPE pumping system, disruption of the blood-retinal barrier, and intraocular inflammation [35]. Among these putative mechanisms, Müller cell hypertrophy and its paracrine repercussions are deemed pivotal, thus attributing the inner nuclear layer (INL) as the primary locus of CME pathogenesis [37–39]. Furthermore, CME were associated with higher vessel density values, as well as thicker choroidal layers and has been linked to circulatory dynamics [37]. It is conceivable that anomalous responses transpiring at diverse loci spanning from the INL to the choroid might have precipitated CME, with the duration of CME potentially exacerbating the deleterious repercussions on photoreceptor density. This remains speculative and necessitates verification through forthcoming histological studies. Considering that the duration of CME may affect photoreceptors, it is desirable to start treatment immediately when CME complicates.

One of the limitations of this study was that it was a single-point observational study. Thus, it remains unclear whether photoreceptor density decreases due to CME, or whether CME occurs because photoreceptor density has decreased, necessitating follow-up studies involving longitudinal analyses. Nonetheless, the fact that we were able to perform AO on each generation allowed us to consider the effects of CME. Our results suggest that CME should be treated as early as possible to reduce the likelihood of photoreceptor damage and emphasize the importance of follow-up assessment in patients with RP. For these patients, AO imaging may be a valuable and particularly useful tool.

In conclusion, our analysis revealed deterioration of photoreceptor density and regularity in patients with RP complicated by CME. As such, AO technology may become an important tool for the assessment of photoreceptor characteristics in the future.

## Supporting information

**S1 Fig. Prevalence of gene mutation in retinitis pigmentosa patients.** (a) A total of 54 eyes of 42 patients. (b) CME-positive group consisting of 30 eyes of 18 patients. (c) CME-negative group consisting of 24 eyes of 24 patients.
(PDF)

**S2 Fig. Violin plot comparison of photoreceptor density in groups with and without CME.**
(a) One eye per case. (b) Two eyes per case.
(PDF)

**S3 Fig. Violin plot comparison of photoreceptor regularity in groups with and without CME.** (a) One eye per case. (b) Two eyes per case.
(PDF)

**S1 Table. List of gene mutations associated with retinitis pigmentosa.**
(DOCX)

## Acknowledgments

We thank the genetics team at Kobe City Eye Hospital, Kazusa DNA Research Institute (Chiba, Japan) for genetic analysis and technical support, and all members of the Kobe City Eye Hospital for their ongoing support. We are also grateful to statistics consultation team of Kobe City Medical Center General Hospital for collaboration of this work.

## Author Contributions

**Conceptualization:** Shohei Kitahata, Kiyoko Gocho.

**Data curation:** Shohei Kitahata.

**Formal analysis:** Shohei Kitahata.

**Investigation:** Shohei Kitahata, Naohiro Motozawa, Satoshi Yokota, Midori Yamamoto, Yasuhiko Hirami.

**Methodology:** Shohei Kitahata.

**Project administration:** Shohei Kitahata.

**Supervision:** Kiyoko Gocho, Akiko Maeda, Yasuhiko Hirami, Yasuo Kurimoto, Kazuaki Kadonosono, Masayo Takahashi.

**Writing – original draft:** Shohei Kitahata.

**Writing – review & editing:** Shohei Kitahata, Kiyoko Gocho.

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
