## [Decision Letter · Decision Letter 0]

17 Sep 2023

PONE-D-23-24372Evaluation of Photoreceptor Features in Retinitis Pigmentosa with Cystoid Macular Edema by Using an Adaptive Optics Fundus CameraPLOS ONE

Dear Dr. Gocho,

Thank you for submitting your manuscript to PLOS ONE. After careful consideration, we feel that it has merit but does not fully meet PLOS ONE’s publication criteria as it currently stands. Therefore, we invite you to submit a revised version of the manuscript that addresses the points raised during the review process. Please submit your revised manuscript by Nov 01 2023 11:59PM. If you will need more time than this to complete your revisions, please reply to this message or contact the journal office at plosone@plos.org. Please include the following items when submitting your revised manuscript:A rebuttal letter that responds to each point raised by the academic editor and reviewer(s). You should upload this letter as a separate file labeled 'Response to Reviewers'.A marked-up copy of your manuscript that highlights changes made to the original version. You should upload this as a separate file labeled 'Revised Manuscript with Track Changes'.An unmarked version of your revised paper without tracked changes. You should upload this as a separate file labeled 'Manuscript'.

We look forward to receiving your revised manuscript.

Kind regards,

Jiro Kogo

Academic Editor

PLOS ONE

Journal Requirements:

2. Please include complete captions for your Supporting Information files at the end of your manuscript, and update any in-text citations to match accordingly. Please see our Supporting Information guidelines for more information: http://journals.plos.org/plosone/s/supporting-information.

Reviewers' comments:

Reviewer's Responses to Questions

**Comments to the Author**

1. Is the manuscript technically sound, and do the data support the conclusions?

Reviewer #1: Partly

Reviewer #2: Partly

2. Has the statistical analysis been performed appropriately and rigorously? 

Reviewer #1: N/A

Reviewer #2: Yes

3. Have the authors made all data underlying the findings in their manuscript fully available?

Reviewer #1: No

Reviewer #2: Yes

4. Is the manuscript presented in an intelligible fashion and written in standard English?

Reviewer #1: Yes

Reviewer #2: Yes

5. Review Comments to the Author

Reviewer #1: The authors used AO-SLO to evaluate photoreceptors in RP-associated CME. CME is a relatively common complication in RP, and investigating the etiology is clinically important. However, there are some concerns for the interpretation of the result. Specific comments are listed below.

1. Line 93, All participants provided written informed consent. However, the study is retrospective and the written informed consent seems not mandatory. In addition, the study was permitted in 2020, and the data between 2015 and 2021 were collected. Was the informed consent obtained for another study such as genetic testing? Please specify.

2. Line 95 visual field constriction not construction

3. Line 151, Although the authors state that the interpretation is based on ACMG guidelines, the presented criteria seem not follow the guideline. Specifically, null variant can be categorized as PVS1 but PVS1 without a strong or moderate category is considered VUS. Allele frequency of 5% is too high as a cut-off value. Even if the cut-off were accepted, the assigned category is PM2 and requires PVS, PS, or another 2 PM category to be pathogenic/likely pathogenic. Positive results of in-silico analysis are PP3 and need even more evidence.

4. Please specify the identified variants in addition to gene names.

5. Line 223, The authors did not find a positive correlation between EZ/IZ length and photoreceptor cell counts in CME-positive group. However, the p-value is 0.10, and the non-significance may be due to insufficient sample size. Discussing the difference between significant and non-significant p-value of the comparison should be avoided.

6. Line 312, In contrast, the authors discussed CME may disrupt EZ/IZ structure and impair retinal function despite that there was no significant difference in EZ/IZ between CME positive and negative groups (Table 3). Visual acuity looks even better in the CME-positive group. The authors should reconsider the interpretation of the results.

7. Related to the above comment, the question is whether the CME-negative group is an appropriate control for the CME-positive group. How did the authors select the CME-negative cases?

8. Is it possible that the presence of CME affects the AO-SLO imaging and results in lower cell density?

9. As the authors introduced, studies on AO-SLO findings in RP is limited. Nevertheless, the citation list seems not comprehensive. Previous studies on RP would be more relevant to the present study than MacTel type 2 or GA papers.

Reviewer #2: In this study, the authors evaluated the photoreceptor features in retinitis pigmentosa with cystoid macular edema using an adaptive optics fundus camera. They use many Tables and Figures for various analysis, but I recommend that they need to be clear and compact this text, tables, and figures.

1. In this study, CME-positive group include the 13 eyes in current and the 5 eyes in previous. Does the presence of CME affect the photoreceptor mosaic imaging due to adaptive optics fundus camera image quality issues? Although the number of cases is small, is there any difference between the 13 eyes with current CME and the 5 eyes with previous CME?

2. In this data, the integrity of the ellipsoid zone and interdigitation zone is not significantly different between CME-positive group and CME-negative group, but there is a significant difference in photoreceptor density and regularity of AO-camera. Visual acuity is also not significantly different between the two groups, so could this reduction in photoreceptor density and regularity just not reflect well due to the presence of CME? Is there any study that shows that CME causes faster progression of retinitis pigmentosa or faster loss of central vision? The authors need to be considered more.

3. Figure 5-7: Please note the P-values and correlation coefficients in the figure or figure legend.

4. Table 4-5: Please clearly indicate in the table title that the data is only for the CME-positive group.

5. Table 4-5: How did the authors define disease duration?

6. PLOS authors have the option to publish the peer review history of their article (what does this mean?). If published, this will include your full peer review and any attached files.

Reviewer #1: No

Reviewer #2: No

---

## [Author Response · Author response to Decision Letter 0]

4 Nov 2023

Thank you for inviting us to submit a revised manuscript. We appreciate the time and effort dedicated to providing insightful feedback to strengthen our manuscript. We have worked to address each concern by additional analysis and revising the text for clarity and added results. Thus, with great pleasure we submit our revised manuscript for further consideration. We have incorporated changes that reflect the detailed suggestions you have graciously provided. We provided an explanation in good faith in response to the reviewers we submitted this time.

---

## [Decision Letter · Decision Letter 1]

20 Nov 2023

PONE-D-23-24372R1Evaluation of Photoreceptor Features in Retinitis Pigmentosa with Cystoid Macular Edema by Using an Adaptive Optics Fundus CameraPLOS ONE

Dear Dr. Gocho,

Thank you for submitting your manuscript to PLOS ONE. After careful consideration, we feel that it has merit but does not fully meet PLOS ONE’s publication criteria as it currently stands. Therefore, we invite you to submit a revised version of the manuscript that addresses the points raised during the review process.

We look forward to receiving your revised manuscript.

Kind regards,

Jiro Kogo

Academic Editor

PLOS ONE

Journal Requirements:

Reviewers' comments:

Reviewer's Responses to Questions

**Comments to the Author**

1. If the authors have adequately addressed your comments raised in a previous round of review and you feel that this manuscript is now acceptable for publication, you may indicate that here to bypass the “Comments to the Author” section, enter your conflict of interest statement in the “Confidential to Editor” section, and submit your "Accept" recommendation.

Reviewer #1: All comments have been addressed

Reviewer #2: All comments have been addressed

2. Is the manuscript technically sound, and do the data support the conclusions?

Reviewer #1: Yes

Reviewer #2: Partly

3. Has the statistical analysis been performed appropriately and rigorously? 

Reviewer #1: Yes

Reviewer #2: Yes

4. Have the authors made all data underlying the findings in their manuscript fully available?

Reviewer #1: No

Reviewer #2: Yes

5. Is the manuscript presented in an intelligible fashion and written in standard English?

Reviewer #1: Yes

Reviewer #2: Yes

6. Review Comments to the Author

Reviewer #1: The authors addressed the raised concerns. The comment AO imaging may detect photoreceptor loss that cannot be seen as EZ/IZ shortening is interesting, but it is not clearly stated in the manuscript. The authors may want to emphasize the significance of AO imaging.

Reviewer #2: 1. Line 123 AO analysis; As pointed out in the first review, I am most concerned that the presence of edema itself will affect the AO camera imaging and cause the cone density to be measured lower. The authors state that “For each photoreceptor analysis examination, we captured the scans of the four perifoveal areas of the retina, as the regions of interest (ROI): superior, inferior, temporal, and nasal, 1.5° to 2° from the center of the fovea with a standardized sampling window of 62 × 62 µm sampling window size”. On the other hand, they also state that “In order to avoid the influence of CME on the measurement results of the AO examination, measurements were obtained from regions in which CME had occurred previously but not in areas with current edema in the OCT examination”.

⇒ Were not measurements taken at four fixed locations? If CME was present, did you exclude data from that measurement site? I recommend adding a typical example to Figure so that we can see how the measurements were taken in cases where CME is present.

2. The term "disease duration" or “duration” in the table 4 and in the text should be clearly stated as duration of CME, which is misleading as duration of retinitis pigmentosa.

7. PLOS authors have the option to publish the peer review history of their article (what does this mean?). If published, this will include your full peer review and any attached files.

Reviewer #1: No

Reviewer #2: No

---

## [Author Response · Author response to Decision Letter 1]

10 Dec 2023

Thank you for inviting us to submit a revised manuscript. We appreciate the time and effort dedicated to providing insightful feedback to strengthen our manuscript. We have worked to address each concern by additional analysis and revising the text for clarity and added results. We are again sincerely responding to the points raised by the reviewers.

---

## [Editor Report · Decision Letter 2]

15 Dec 2023

Evaluation of Photoreceptor Features in Retinitis Pigmentosa with Cystoid Macular Edema by Using an Adaptive Optics Fundus Camera

PONE-D-23-24372R2

Dear Dr. Gocho

We’re pleased to inform you that your manuscript has been judged scientifically suitable for publication and will be formally accepted for publication once it meets all outstanding technical requirements.

Kind regards,

Jiro Kogo

Academic Editor

PLOS ONE

---

## [Editor Report · Acceptance letter]

20 Dec 2023

PONE-D-23-24372R2 

PLOS ONE

Dear Dr. Gocho, 

I'm pleased to inform you that your manuscript has been deemed suitable for publication in PLOS ONE. Congratulations! Your manuscript is now being handed over to our production team.

Kind regards, 

on behalf of

Dr. Jiro Kogo 

Academic Editor

PLOS ONE